# Integrated Approach for the Optimization of the Sustainable Extraction of Polyphenols from a South American Abundant Edible Plant: *Neltuma ruscifolia*

**DOI:** 10.3390/foods14172927

**Published:** 2025-08-22

**Authors:** Giuliana S. Seling, Roy C. Rivero, Camila V. Sisi, Verónica M. Busch, M. Pilar Buera

**Affiliations:** 1Facultad de Bromatología, Universidad Nacional de Entre Ríos, Gualeguaychú 2820, Entre Ríos, Argentina; camila.sisi@uner.edu.ar (C.V.S.); giuliana.seling@uner.edu.ar (G.S.S.); roy.rivero@uner.edu.ar (R.C.R.); 2Instituto de Tecnología de Alimentos y Procesos Químicos—ITAPROQ (CONICET-UBA), Ciudad Autónoma de Buenos Aires 1428, Argentina; 3Instituto de Ciencia y Tecnología de los Alimentos de Entre Ríos—ICTAER (CONICET-UNER), Gualeguaychú 2820, Entre Ríos, Argentina; 4Departamento de Industrias, Facultad de Ciencias Exactas y Naturales, Universidad de Buenos Aires, Ciudad Autónoma de Buenos Aires 1428, Argentina

**Keywords:** experimental design, polyphenols, antioxidant capacity, flavonoids, bioactive extracts

## Abstract

The pods from *Neltuma ruscifolia* (*vinal*), an underutilized species, are rich in bioactive functional compounds. However, the extraction procedures to obtain the highest proportion of these compounds, considering sustainability aspects, have not been optimized. This study aimed to optimize and compare three affordable extraction methods—dynamic maceration (DME), ultrasound-assisted extraction (UE), and microwave-assisted extraction (ME)—to obtain enriched extracts. The effects of temperature, ethanol-to-water ratio in the solvent, extraction time, and frequency (for ME) were evaluated using a Box–Behnken design and response surface methodology to optimize total polyphenolic content (TPC), total flavonoids (TF), and antioxidant capacity (DPPH). Energy consumption and carbon footprints were also assessed, and phenolic compounds in the optimized extracts were identified by HPLC. The ethanol-to-water ratio emerged as the most influential factor, showing synergistic effects with both time and temperature, enabling optimal yields at intermediate ethanol concentrations. Gallic acid, rutin, and theobromine were found to be the most abundant components, followed by cinnamic, caffeic, and chlorogenic acids. Although UE exhibited the lowest energy consumption (0.64 ± 0.03 Wh/mg of TPC), the simple and easily implementable DME—optimized at 40 min, 50 °C, and 42% ethanol—proved to be the most efficient method, combining high extractive performance (TPC 1432 mg GAE/100 g Dw), reduced solvent use, and intermediate energy efficiency (1.84 Wh/mg of TPC). These findings highlight the potential of *vinal* as a natural source of bioactive ingredients obtained through simple and cost-effective techniques adaptable to small producers while underscoring the value of experimental design in optimizing sustainable extraction technologies and elucidating the interactions between key processing factors.

## 1. Introduction

*Neltuma ruscifolia* (Griseb.) C.E. Hughes & G.P. Lewis, formerly known as *Prosopis ruscifolia*, is a woody leguminous species native to South America, particularly from the arid and semi-arid regions of Argentina, Paraguay, and Bolivia. It is currently classified as a Neglected and Underutilized Species (NUS) owing to its ecological resilience, multiple uses, and potential to contribute to sustainable food systems and to rural and economically disadvantaged regions [1,2]. Similar to other species within the genus *Prosopis*, *N. ruscifolia* has a long history of utilization by Indigenous and rural communities, who have traditionally consumed its pods as food and forage and used its wood as a source of fuel and for construction purposes [3,4]. These species have been valued since pre-Columbian times for their nutritional content and adaptability to harsh environments, representing an important cultural and subsistence resource. Nevertheless, *N. ruscifolia* remains largely overlooked in agricultural research, development policies, and commercial value chains [5]. Promoting the valorization of *N. ruscifolia* supports global efforts to enhance agrobiodiversity, food security, and rural livelihoods by integrating NUS into sustainable food systems [6,7].

Several other South American *Prosopis* spp. have been the subject of phytochemical and pharmacological investigations, highlighting their potential as sources of functional food ingredients and natural therapeutics. For instance, *Prosopis* alba and *Prosopis* nigra, both native to the Gran Chaco region, have demonstrated high phenolic content and strong antioxidant capacity in their pods and flour extracts [8,9]. Additionally, *P. alba* leaf and bark extracts have shown anti-inflammatory and cytoprotective effects in in vitro studies [10], while *P. nigra* has exhibited antimicrobial activity against foodborne pathogens [11]. The known bioactivity of the leaves, including antioxidant, antimicrobial, and anti-inflammatory properties, has been documented in related *Prosopis* spp. [12,13], and recent studies confirm similar properties in the fruits of *N. ruscifolia* [14]. Seling et al. [14] observed that the antioxidant, antiglycating, colorant, and technofunctional properties of extracts from *Neltuma ruscifolia* pods were primarily determined by the selected milling fraction and by the extraction method in a secondary place, which influenced both the yield and the composition of bioactive compounds, having identified chrysin, rutin, kaempferol, and cinnamic, coumaric, protocatechuic, ellagic, and caffeic acids [14].

Response surface methodology (RSM) is a powerful statistical tool that enables the modeling and optimization of bioactive compound extraction processes through the use of experimental designs (such as Box–Behnken), empirical models, and desirability functions to simultaneously address multiple responses. This approach reduces the number of experiments required and allows the investigation of both main effects and factor interactions, thus achieving optimal extraction conditions [15]. In this context, the valorization of *N. ruscifolia* through the development and optimization of bioactive extracts appears particularly promising to add value and provide potential economic resources for vulnerable areas or small producers.

Moreover, the adoption of affordable green extraction technologies, such as ultrasound-assisted extraction (UE) and microwave-assisted extraction (ME), provides environmentally friendly and efficient alternatives to conventional methods. At the cellular level, ME rapidly heats water and other dipoles inside cells, generating internal pressure that ruptures cell walls and membranes, releasing bioactive compounds. On the other hand, UE produces cavitation—microbubble formation and implosion—creating microjets and shock waves that damage cellular structures and enhance solvent permeability. Mechanical agitation, in contrast, only improves mass transfer via convection and diffusion, without directly disrupting cells, typically resulting in lower extraction efficiency. These techniques can improve the yield and selectivity of phenolic and flavonoid compounds while minimizing solvent consumption, processing time, and energy use [16,17]. Their application aligns with the principles of green chemistry and sustainable bioprocessing and has already proven successful in the extraction of bioactives from various arid-region legumes and underutilized plants [18]. In this way, all three aspects of sustainable development (economic, environmental, and social) are considered.

This study aimed to optimize the extraction process for the sustainable recovery of bioactive compounds from *N. ruscifolia* pod powder. Three simple, affordable, and sustainable extraction techniques were compared using response surface methodology (RSM) with a Box–Behnken design, evaluating recovery efficiency, energy consumption, and carbon footprints.

## 2. Materials and Methods

### 2.1. Conditioning of Vinal Pods

The fruits of *Neltuma ruscifolia* (*vinal* pods) were harvested at their optimal ripening stage in Santiago del Estero, Argentina, during November–December of 2023. Figure 1 shows *Neltuma ruscifolia* pods, branches, and flowers with the area of occurrence in Argentina, South America. The pod powder was obtained using Ojeda et al.’s [19] methodology. After washing, the pods were disinfected in a 5% sodium hypochlorite solution and subsequently dried at 50 °C for 3.5 h using a tray dehydrator (FA 10-MZ; COBOS, Buenos Aires, Argentina). The dried material was then ground using a mill (HC-1000 Y, Arcano, Fuzhou, China) and sieved through stainless steel meshes (A.S.T.M. No. 5, 7, 10, and 20; Zonytest, Buenos Aires, Argentina) to obtain a particle size smaller than 840 μm [19]. The resulting powder was stored at −20 °C until analysis.

### 2.2. Preparation of Ethanolic Extracts

The general procedures for extract preparation were adapted from Rivero et al. [20], with some modifications. In each case, 5 g of *vinal* powder was suspended in 95 g of an ethanol–water solution. Three extraction techniques were evaluated as follows: dynamic maceration extraction (DME) using a magnetic stirrer (Precytec AE-29, Buenos Aires, Argentina) at 200 rpm with temperature control via thermostatic bath (FALC SB 15, Treviglio, Italy); microwave-assisted extraction (ME) with agitation at 200 rpm (Bioevopeak MWO-J1 equipment, Jinan, China); and ultrasound-assisted extraction (UE) (Sonics Vibra Cell, Newtown, CT, USA) with constant agitation at 200 rpm (Precytec AE-29). The specific conditions for each method are detailed in the experimental design. After extraction, the mixtures were centrifuged at 2500 rpm for 10 min; the supernatants were collected, stored in amber glass bottles, and kept at −20 °C until analysis. For comparative purposes, traditional maceration extraction (TME) was also included, consisting of soaking the material at room temperature for 15 days.

### 2.3. Experimental Design for RSM Optimization

The extraction process was optimized using a Box–Behnken response surface design with Design Expert 11 software. The evaluated parameters were as follows: for DME, ethanol concentration (ET) from 0% to 100%, extraction temperature (T) from 20 °C to 50 °C, and agitation time (t) from 10 to 40 min; for ME, ET (0–100%), T (20–50 °C), t (10–40 min), and heating intensity level (IN) from 160 to 800 W; and for UE, ET (0–100%), t (1–15 min), and amplitude (A) from 20% to 100%. The response variables analyzed in all experiments included antioxidant capacity (measured by DPPH), total flavonoid content, total polyphenolic compounds (TPC), and relative energy efficiency.

The experimental design applied to the three extraction methods (DME, UE, ME) was fitted to quadratic models to describe the interactions among the factors influencing the extraction of bioactive compounds from the matrix. Mathematical models were constructed based on the experimental results (Appendix A), and the data were fitted to second-order equations (Appendix A). The significance of the factors and the statistical parameters associated with the model fit were evaluated through an analysis of variance (ANOVA).

### 2.4. Characterisation of the Response Variables

#### 2.4.1. Antioxidant Capacity (DPPH and ABTS Assays)

The antioxidant capacity (AC) was evaluated using the Trolox Equivalent Antioxidant Capacity (TEAC) assay, where 25 μL of each extract was mixed with 1975 μL of an ethanolic DPPH solution (Sigma Aldrich^®^, St. Louis, MO, USA). After 30 min of incubation in the dark, the absorbance was measured at 515 nm using a Trolox standard curve (0.2–1.2 mg/mL), following the methodology described by Rolandelli et al. [21]. The results were expressed as mmol of Trolox g of dry solids of the original powder (Dw).

Additionally, the antioxidant capacity was determined using the TEAC assay based on the 2,2′-azino-bis(3-ethylbenzothiazoline-6-sulfonic acid) radical cation (ABTS•^+^). A 7 mM ABTS•^+^ solution (Sigma Aldrich^®^) was prepared, and the absorbance was measured at 734 nm using the same Trolox standard curve employed for the AC assay, according to the method described by Rivero et al. [22]. The results were expressed as mmol of Trolox (TEAC) per 100 g Dw.

#### 2.4.2. Total Polyphenolic Content (TPC)

The Folin–Ciocalteu method was used to determine TPC. Absorbance was measured at 765 nm, and a standard curve of gallic acid (Merck, Darmstadt, Germany) was prepared with six points ranging from 0 to 0.5 mg/mL [23]. Results were expressed as mg of gallic acid equivalents (GAE) per 100 mL of extract according to Seling et al. [14] and per 100 g Dw.

#### 2.4.3. Total Flavonoids (TF)

For the determination, a quercetin (Sigma Aldrich^®^) standard curve was prepared with six points, ranging from 0 to 8 ppm. For the measurement, a 0.1 mL aliquot of the *vinal* extract was reacted with 0.05 mL of a 2% aluminum chloride solution and 2.3 mL of ethanol. The reaction was incubated at 25 °C in darkness. Subsequently, the samples and the standard curve were measured at a wavelength of 415 nm. The results were expressed as milligrams of quercetin per 100 mL of the extract (mg EQ/100 mL) [24] and per 100 g Dw.

The AC, TPC, and TF determinations were carried out using a Jenway 6505 ultraviolet–visible spectrophotometer (Burlington, NJ, USA).

#### 2.4.4. Relative Energy Efficiency

The energy efficiency of each extraction method was estimated by considering the electrical energy consumed during the process and the TPC obtained under the studied conditions. The energy consumption (E) was calculated according to Equation (1):*E*(*Wh*) = *P*(*W*) × *t*(*h*)(1)
where

P is the nominal power of the equipment used (in watts);

t is the effective extraction time (in hours), based on the optimized experimental conditions for each technique.

The P values were obtained from the equipment specifications provided by the manufacturer or measured using an electricity consumption meter, and the t values were derived from the experimental design.

The relative energy efficiency was calculated according to Equation (2), expressed as the amount of energy consumed per milligram of gallic acid equivalents (mg GAE) of TPC extracted:(2)$$\mathrm{Energy}\;\mathrm{efficiency}\;(\mathrm{Wh}/\mathrm{mgGAE})\;=\;\frac{E(Wh)}{TPC(mgGAE)}$$

This parameter allowed for an objective comparison between the extraction methods, taking into account the energy consumption required per unit of total phenolic compounds obtained.

#### 2.4.5. Total and Relative Carbon Footprint

The total carbon footprint associated with each extraction method was estimated by considering the electrical energy consumed during the process and the carbon dioxide emission factor for the regional energy matrix. The energy consumption (E) was calculated according to Equation (3):*E* = *P* × *T*(3)
where

P is the nominal power of the equipment used (in kilowatts, kW);

T is the effective extraction time (in hours), based on the optimized conditions for each technique.

The total carbon footprint (CF) was calculated by multiplying the energy consumption by the carbon dioxide emission factor (F), according to Equation (4):*CF* = *E* × *F*(4)
where

F is the carbon dioxide emission factor for the regional energy matrix (kg CO_2_/kWh).

This approach to estimating the total carbon footprint has been widely applied in previous studies to assess the environmental impact of laboratory processes [25,26].

To compare the environmental efficiency of the extraction methods in terms of functional compound production, the relative carbon footprint (RCF) was calculated, expressed as the amount of equivalent carbon dioxide emissions per milligram of TPC. This calculation was performed by Equation (5).
(5)$$\mathrm{RCF}=\frac{CF}{TPC}$$where *TPC* represents the amount of total polyphenolic content extracted (mg). This parameter allowed for an objective evaluation of the sustainability of the analyzed methods, considering the environmental footprint per unit of functional compound obtained.

### 2.5. Polyphenol Profile by HPLC

Serum samples were aliquoted and filtered using a 0.45 µm syringe filter. Then, 20 µL of the samples was injected for chromatographic analysis. The chromatographic run was performed according to [27] with slight modifications. A Waters 1525 HPLC system (Milford, CT, USA) equipped with a binary pump and a 2996 photodiode array detector (PDA) (Waters, Milford, USA) was used. A Restek Roc C18 column (125 × 4.6 mm) was employed. Mobile phase A consisted of acetonitrile–methanol (50:50), while mobile phase B consisted of water and phosphoric acid (99:1). The gradient program was optimized by changing the percentage of mobile phases and was set as follows: Time (min)/mobile phase A:B (%): T0/5:95, T3/5:95, T7/20:80, T15/20:80, T18/30:70, T30/30:70, and T35/5:95, at a flow rate of 1 mL/min. For the identification and quantification of polyphenolic compounds, an external standard method was employed. Identification was based on the retention time and absorption spectrum of each compound, and quantification was performed by comparing the area under the curve for each identified peak with the areas of external standards of known concentration.

### 2.6. Comparing Extraction Methods

Once the optimal extraction conditions for each of the studied methods were established, the corresponding extracts were obtained under those conditions. The responses were then reanalyzed to verify the predictive level of the designs and to select the most efficient method for extracting bioactive compounds. Additionally, the methods were compared with the traditional extraction technique to determine the most effective alternative for producing extracts rich in functional compounds.

### 2.7. Statistical Analysis

All determinations were performed in triplicate, and mean values and standard deviations were reported. Statistical analysis of the results was performed through ANOVA for a level of significance (α) of 0.05, followed by the LSD Fisher post hoc test to identify significant differences among the systems. Statgraphics Centurion XV software (V 2.15.06, 2007, Statpoint Technologies, Inc., Herndon, VA, USA) and InfoStat 4 (Grupo InfoStat, FCA, Universidad Nacional de Córdoba, Córdoba, Argentina) were used. Multivariate analysis was conducted using the multivariate and classification analysis module. The data were processed using the InfoStat-Statistical software, student version 2017.

## 3. Results and Discussion

### 3.1. Optimized Conditions for Extraction

#### 3.1.1. Experimental Data

The experimental results were obtained using a Box–Behnken design, employing random factor combinations with 15 runs for the DME and UE techniques and 27 runs for ME, as detailed in Appendix A. This table compiles the analyzed factors and responses, the type and significance of each modeled fit, and the R^2^ and adjusted R^2^ values for each response model.

Regarding AC, the DME method yielded values ranging from 3.3 to 10.7 mmol Trolox/100 g Dw. The UE method showed values between 1.3 and 7.7, while the ME method recorded values from 0.7 to 7.7. All three methods effectively extracted DPPH-scavenging compounds, likely including flavonoids, phenolic acids, and tannins [28]. Among the tested methods, DME achieved the highest extraction of antioxidant compounds, exhibiting the strongest correlation with the other evaluated variables (TPC and TF).

The TF content (mg QE/100 g Dw) obtained by DME ranged from 18.6 to 166.4; for UE, the values varied from 14.9 to 118.2; and for ME, the values were between 6.1 and 108.8. Regarding TPC (mg GAE/100 g Dw), DME yielded values from 260.6 to 1377.3, UE showed a range of 144.5 to 1081.7, whereas ME produced values between 177.4 and 1134.2. These results demonstrate that DME achieved the highest concentrations of functional compounds and comparatively higher minimum values. The R^2^ values demonstrated a strong correlation between the extraction parameters and the recovery of TPC and TF, as also highlighted by Mona Hamwi et al. [29], who reported a correlation of 0.998.

In general, previous studies on other *Neltuma* or *Prosopis* spp. have reported a diverse polyphenolic profile, mainly composed of flavonoids, tannins, anthocyanins, gallocatechin, coumaric acid, morin, rutin, catechin, gallic acid, naringenin, and epicatechin, among others, which account for more than 98% of their AC [29,30].

#### 3.1.2. Factor Effect on Responses

The experimental design applied to the three extraction methods (DME, UE, ME) was fitted to quadratic models, reflecting the complexity of the interactions among the factors involved in the extraction of bioactive compounds from the matrix (Appendix A). As shown in Appendix A, the ANOVA results confirmed that all models were statistically significant (*p* ≤ 0.0001), except for the TF model under UE. Additionally, no significant lack of fit was detected (*p* ≥ 0.0583), and the adequate precision was satisfactory in all cases (greater than 11.37), a parameter that measures the signal-to-noise ratio and is considered acceptable when exceeded [31].

Therefore, the predictive models obtained are valid for exploring the design space, except for TF under UE and ME. Overall, the models developed for DME exhibited superior statistical performance. Based on the values in Appendix A and the mathematical models in Appendix A, three-dimensional plots were generated to visualize the individual and synergistic effects of the factors on the target responses.

a.Dynamic Maceration

The analysis of AC was fitted to a statistically significant quadratic model, showing high agreement between the experimental and predicted values. According to the ANOVA, the individual effects of t, T, and ET, as well as the interactions t·T and t·ET, were significant. The quadratic terms t^2^ and ET^2^ were also significant, indicating a nonlinear response, particularly related to ethanol and the complex interactions among the factors.

In the t–T interaction (Figure 2Aa), it was observed that at higher temperatures, an increase in extraction time enhanced AC, whereas at lower temperatures, this effect was negligible. This corresponds to the positive coefficient of the t·T interaction and the moderate curvature of the quadratic term t^2^. In the ET–t interaction, a bell-shaped profile was identified, with an AC peak between 40% and 60% ethanol, decreasing outside this range (Figure 2Ab). This response is explained by the significance of the negative quadratic term ET^2^. Although time had a limited effect, its combination with intermediate ethanol concentrations improved extraction. For the ET–T interaction, a similar trend was observed, with a moderate positive effect of T within the optimal ET range and a pronounced decrease at higher concentrations, consistent with the low influence of the T·ET interaction (Figure 2Ac).

For TF, the ANOVA indicated significant individual effects of T and ET, as well as a notable t·T interaction (Figure 2B). The curvature associated with ET, explained by the quadratic term ET^2^, revealed a parabolic response with a peak at intermediate ethanol concentrations, similar to that observed for AC. In the t–T surface (Figure 2Ba), it was confirmed that the simultaneous increase in T and t enhanced TF, whereas at lower temperatures, the effect of t was minimal. This trend was consistent with the positive coefficients of T, t, and their interaction. In the ET–t and ET–T surfaces (Figure 2Bb,Bc), respectively), a bell-shaped profile was observed, with reduced TF extraction at the extremes of ethanol concentration.

Regarding TPC, the individual effects of T and ET, along with the t·T and t·ET interactions, were statistically significant, as were the quadratic terms T^2^ and ET^2^ (Figure 2C). In the t–T interaction (Figure 2Ca), high temperatures combined with prolonged times favored TPC extraction, while at low temperatures, the effect of t was limited. This pattern mirrored that observed in AC and TF, though with lower intensity. In the ET–t surface (Figure 2Cb), a peak was identified around 40% ethanol, with a pronounced decline toward concentrations near 100%, consistent with the strong curvature of the ET^2^ term. Time had a moderate influence, showing a slight increase in TPC with longer times. In the ET–T interaction (Figure 2Cc), intermediate ethanol concentrations combined with high temperature promoted greater extraction, whereas high ethanol concentration significantly reduced the response, in line with the contributions of the quadratic terms of ET and T.

b.Ultrasound-assisted extraction

The response surface analysis for AC (Figure 3A) showed high agreement with the fitted quadratic model, which was statistically significant according to ANOVA. The individual effects of ET, A, and t were significant, all exhibiting negative influences on the response. The interactions A·t, A·ET, and ET·t were also significant, along with the three quadratic terms (t^2^, A^2^, ET^2^). The ET–A interaction (Figure 3Aa) displayed a typical parabolic behavior, with a maximum between 40% and 60% ethanol, followed by a sharp decline beyond this range, particularly at high ethanol concentrations. Although A showed a negative linear effect, its positive quadratic term indicated a modulatory role. Low and high amplitudes (20% and 100%) were associated with higher AC, with a minimum around 60%. In the t–A interaction (Figure 3Ab), AC values increased with sonication time, especially at low amplitudes. The positive effect of t was also evident in the t–ET interaction (Figure 3Ac), where maximum AC values were achieved with 40% ethanol and prolonged sonication times, highlighting the importance of both variables in the extraction process.

For TF (Figure 3B), the quadratic model was not globally significant according to ANOVA. However, the ET^2^ term was significant, indicating a clear parabolic response to ethanol content. The ET–A surface (Figure 3Ba) showed a pattern similar to AC, with maximum TF values between 40% and 60% ethanol. The negative ET^2^ coefficient confirmed this bell-shaped profile, with reduced extraction at extreme ethanol concentrations. Unlike AC, the effect of t was more pronounced at high amplitudes, consistent with the positive effect of t and the negative A·t interaction. In the t–A interaction (Figure 3Bb), TF increased with time, peaking around 11 min, particularly at high amplitudes. A also exhibited a nonlinear effect, with improved extraction at both ends of the amplitude range (20% and 100%), especially at 100%, which may be attributed to more efficient extraction. The t–ET interaction (Figure 3Bc) confirmed optimal TF extraction between 40% and 60% ethanol, with lower values at 100% ethanol and 1 min of sonication.

For TPC (Figure 3C), the quadratic model showed individual effects of ET (negative) and t (positive). The A·ET and ET·t interactions were also significant, and the ET^2^ term highlighted the combined linear and nonlinear behavior. In the ET–A interaction (Figure 3Ca), the extraction pattern resembled that of AC, with a maximum between 40% and 60% ethanol and a sharp decline at higher concentrations. A had a minor effect, with slight improvements at the extremes, consistent with the positive A^2^ term. In the t–A interaction (Figure 3Cb), the TPC values increased with sonication time, particularly at low amplitudes, aligning with the positive coefficient of t and the negative A·t interaction. The t–ET surface (Figure 3Cc) showed optimal extraction around 40% ethanol, with a positive trend as time increased, supported by the significant effect of t and the negative ET·t interaction.

c.Microwave-assisted extraction

The response surface analysis for AC (Figure 4A) showed that the linear terms for T and t exhibited positive effects, consistent with the surface plots, where the increase in both factors enhanced AC (Figure 4Aa). This effect was more pronounced at high temperatures, suggesting a synergistic relationship, although the t·T interaction was not significant. The negative quadratic term t^2^ indicated a slight downward curvature, with a maximum around 25–28 min in interactions involving t. The quadratic term for ET was the most prominent (Figure 4Ab), producing a bell-shaped curve with a minimum at the range extremes (particularly at 100% ET), indicating an optimal ethanol concentration of 40–60% to maximize AC. The influence of IN was limited, both linearly and quadratically, reflected in minimal AC variation, except under conditions of high T or specific ET levels (Figure 4Ac). The surface plots showed that at low temperatures, IN had negligible effects, but at high temperatures, higher AC values were associated with low intensities, consistent with negative coefficients in some interaction terms.

For TF (Figure 4B), the fitted quadratic model showed that the linear factors T, IN, and ET were significant. Also, it could be noted that the t·T interaction was the only significant interaction, and the quadratic terms t^2^, T^2^, and ET^2^ indicated notable nonlinear behaviors. In the t·T interaction (Figure 4Ba), TF extraction increased with time, peaking at low temperatures, while shorter times were required at high temperatures, suggesting thermal degradation, aligned with the t·T and T^2^ coefficients. The t·IN interaction (Figure 4Bb) showed that at longer times, high intensities reduced TF, reflected in the t·IN and IN^2^ coefficients. The ET–t combination (Figure 4Bc) displayed a bell-shaped curve, with a maximum near 50% ethanol, supported by t·ET and ET^2^. In the T·IN interaction (Figure 4Bd), higher TF values were obtained with low intensity at high temperatures, consistent with the T·IN coefficients. The T·ET interaction (Figure 4Be) showed a parabolic behavior, with a maximum around 50% ethanol, reflecting synergy between T and ET, as indicated by T·ET and ET^2^. The ET·IN interaction (Figure 4Bf) highlighted ET’s importance, while IN had a minor effect, consistent with corresponding coefficients.

For TPC (Figure 4C), the fitted quadratic model showed strong agreement and was significant according to ANOVA. The linear terms for t, T, and ET revealed significant effects: t and T positively influenced extraction, while ET showed a negative linear effect, consistent with maximum extraction at intermediate ethanol concentrations. The quadratic terms t^2^ and ET^2^ were found to be significant, with ET^2^ revealing a parabolic behavior in TPC extraction, peaking at approximately 50% and subsequently decreasing at higher concentrations. The t^2^ term suggested that excessively long times do not further improve yield, indicating a plateau or potential degradation. The t·T interaction (Figure 4Ca) showed a synergistic effect, increasing TPC with longer times and higher temperatures. The t·ET interaction (Figure 4Cc) was negative, reflecting reduced efficiency at extreme ethanol concentrations, even with prolonged extraction. The T·ET and IN·ET interactions (Figure 4Ce,Cf, respectively), both positive, suggested that T can partially offset yield reductions at specific ethanol concentrations, whereas IN exerted only a limited influence, becoming notable primarily when interacting with t (Figure 4Cb).

The choice of extraction solvent and method had previously significantly influenced the yield and profile of bioactive compounds obtained from *Prosopis* spp. pod materials. González-Quijano et al. [32] evaluated different extraction methodologies and demonstrated that aqueous ethanol (EtOH 80%) was effective in extracting vanillic acid, vanillin, ferulic acid, and caffeic acid from *Prosopis* spp. pods. The use of acidified ethanol further improved the release of aglycones such as genistein, although in low concentrations. This confirms that both the nature of the solvent and the pH condition modulate the solubility and stability of phenolic compounds and isoflavones.

Similarly, Díaz-Batalla et al. [33] reported that the application of thermal treatments followed by aqueous ethanol extraction significantly increased the TPC and AC of *Prosopis laevigata* flours. This effect was attributed not only to the enhanced release of bound phenolics but also to the generation of Maillard reaction products, which may act synergistically in scavenging free radicals. Their study also highlighted that the mesocarp-rich fractions yielded higher phenolic content compared to seed or bran fractions, underlining the importance of the anatomical origin of the sample in the extraction outcome.

Pérez et al. [34] confirmed that boiling and aqueous extractions improved the recovery of free phenolics and flavonoids in *Prosopis alba* and *P. nigra* pod flours. Moreover, the selection of methanol–water mixtures facilitated the identification of various polyphenols, such as quercetin-O-glycosides and apigenin-C-glycosides, and influenced their antioxidant and anti-inflammatory activity profiles. Their data showed that *P. alba* presented higher total phenolic and flavonoid contents, while *P. nigra* had higher anthocyanin levels.

González-Cortázar et al. [12] demonstrated that the biological activity of *Prosopis laevigata* pod extracts depends on the solvent used for extraction. The dichloromethane extract showed the strongest anti-inflammatory effect, reducing TPA-induced oedema by 75.96%, while the methanol extract exhibited the highest antimicrobial activity (MIC < 12.5 µg/mL). Fractionation led to the identification of ethyl veratrate, rutin, and quercetin 3-O-glucoside as key bioactive compounds, highlighting the extract’s multifunctional potential.

Appendix A summarizes the findings of these authors, highlighting differences in the raw material and extraction methods. The reported TPC values were either lower than or within the range of those obtained in the present optimization study for *Neltuma ruscifolia*.

In agreement with these previous findings, our results support the premise that the choice of solvent and extraction method is crucial for optimizing the recovery of phenolic compounds (particularly flavonoids and phenolic acids), as solvent polarity influences their capacity to disrupt plant matrices and solubilize hydroxylated aromatic compounds.

#### 3.1.3. Experimental Evaluation of the Optimized Models

After analyzing the response behavior based on the interaction of factors, each extraction method (DME, UE, ME) was optimized to obtain *vinal* extracts under specific optimal conditions, enabling their comparison. The predictive modeling aimed to maximize AC, prioritizing TPC and, to a lesser extent, TF. These response adjustments, along with the expected values derived from mathematical modeling, are presented in the Appendix A.

The determined optimal conditions were as follows: DME at 40 min, 50 °C, and 42% ET; UE at 15 min, 20% A, and 50% ET; and ME at 40 min, 50 °C, 480 W IN, and 50% ET. The predictive capacity of the models was verified through complementary analyses using Design-Expert, with a 95% confidence interval. The comparison between the experimental values obtained under optimal conditions and the theoretical predictions confirmed that all results fell within the established confidence intervals, thereby validating the reliability of the models for predicting response behavior.

For DME, the extracts showed an AC of 9.6 ± 0.2 mmol Trolox/100 g Dw, a TPC of 1432.0 ± 78.8 mg GAE/100 g Dw, and a TF of 131.4 ± 6.6 mg QE/100 g Dw, with percentage differences between experimental and predicted values of 7.2%, 0.5%, and 12.5% for AC, TPC, and TF, respectively. For UE, the values were 5.5 ± 0.9 mmol Trolox/100 g Dw for AC, 1105.8 ± 48.2 mg GAE/100 g Dw for TPC, and 111.7 ± 8.8 mg QE/100 g Dw for TF, with percentage differences of 10.3%, 3.1%, and 15.5%, respectively. For ME, the extracts exhibited an AC of 8.3 ± 0.2 mmol Trolox/100 g Dw, a TPC of 1171.4 ± 89.8 mg GAE/100 g Dw, and a TF of 74.4 ± 4.4 mg QE/100 g Dw, with percentage discrepancies of 10.8%, 4.3%, and 6.9% for AC, TPC, and TF, respectively.

### 3.2. Characterization of the Extracts

#### 3.2.1. Comparison of Functional Characteristics

Figure 5 shows the bioactivity of the optimal extracts obtained from the response surface analyses for each extraction method (Figure 5A: TPC and TF; Figure 5B: AC and TEAC). For TPC (Figure 5A, left), DME presented the highest value, followed by ME and UE, all significantly higher than TME (*p* < 0.05). This outcome suggests that DME maximizes TPC extraction through a favorable combination of temperature, time, and ethanol concentration. For TF (Figure 5A, right), DME again ranked first, while UE and TME did not differ significantly (*p* > 0.05), indicating that ultrasound optimization did not notably improve flavonoid recovery.

In terms of scavenging activity, DME achieved the best performance for AC (Figure 5B, right), followed by ME and UE, all significantly surpassing TME (*p* < 0.05). For TEAC (Figure 5B, left), DME and ME obtained the highest values with no significant difference (*p* > 0.05), while UE and TME showed progressively lower results, with significant differences between all methods (*p* < 0.05). These findings highlight the ability of DME and ME to enhance the extraction of compounds with antiradical properties, likely due to greater matrix disruption and more efficient mass transfer [18].

Taken together, DME proved most effective in maximizing both AC and bioactive compound content, whereas ME and UE provided only partial improvements. It is worth noting that localized high temperatures and free radical generation during ultrasound may degrade certain bioactive compounds, reducing bioactivity [35,36]. Given the extensive literature on ultrasound-assisted extraction, such potential losses in functional quality warrant careful consideration.

#### 3.2.2. Relative Energy Efficiency and Carbon Footprint

The relative energy efficiency of the extraction methods, normalized according to the TPC (expressed as Wh/mg of TPC), showed that the UE method exhibited the highest energy efficiency, with a consumption of 0.64 ± 0.03 Wh/mg of TPC. It was followed by DME, with a value of 1.84 ± 0.10 Wh/mg, while ME proved to be the least energy-efficient, registering a consumption of 5.98 ± 0.46 Wh/mg.

Additionally, the total CF generated by each technique was DME = 0.046 kg CO_2_, UE = 0.013 kg CO_2_, and ME = 0.124 kg CO_2_ (considering the specific energy consumption and the emission factor corresponding to Argentina’s energy matrix of 0.387 kg CO_2_/kWh) [37].

To enable environmental comparison among the methods, the RCF was used (defined as the kilograms of CO_2_ emitted per milligram of TPC extracted). The results indicated that UE also showed the best performance in this regard, with an RCF of 0.00025 kg CO_2_/mg TPC, followed by DME (0.00071 kg CO_2_/mg) and ME, which exhibited the highest value (0.00231 kg CO_2_/mg). These findings clearly highlight the environmental advantage of the UE method in terms of both energy efficiency and climate impact per unit of functional compound obtained.

These findings indicate that, although DME achieved the highest yields in terms of AC and polyphenol content, the UE method was more favorable from an energy efficiency perspective. In contrast, despite achieving acceptable extraction yields, the ME method exhibited a considerably higher energy consumption. Furthermore, the CF analysis reinforced this trend, as UE showed the lowest environmental impact, both in terms of total emissions and emissions per milligram of extracted phenolic compounds.

#### 3.2.3. Polyphenolic Profile of the Optimized Extracts

Table 1 shows the phenolic compounds identified in the optimal extracts by HPLC for each extraction method evaluated. Along with the absolute values, the relative percentages obtained by ME and UE are included, using DME as the reference method (100%).

The results reveal statistically significant differences among the methods. DME achieved the highest concentrations for all the analyzed compounds, particularly for gallic acid (167.8 ppm), theobromine (74.9 ppm), and rutin (114.4 ppm). In contrast, ME and UE exhibited considerably lower efficiency in extracting these key bioactive compounds.

ME yielded concentrations ranging from 23.8% to 68.7% of those obtained by DME, whereas UE produced even lower values, between 1.9% and 51.0%. These results underscore the efficiency gap between assisted methods and DME.

Although some compounds, such as chlorogenic, caffeic, and cinnamic acids, exhibited narrower percentage differences, the overall data confirm that DME consistently outperformed ME and UE in terms of both yield and diversity of phenolic compounds extracted.

The phenolic compounds identified by HPLC in *Neltuma ruscifolia* (*vinal*) extracts exhibit well-documented bioactive properties, thereby supporting their potential application in the food, pharmaceutical, and cosmetic industries. Gallic acid—the predominant compound in DME extracts—demonstrates strong antioxidant, antimicrobial, and anti-inflammatory activities [14,38]. Its presence is pivotal in mitigating oxidative cellular damage, positioning these extracts as promising candidates for the development of functional ingredients.

On the other hand, chlorogenic acid, although found in lower concentrations, has demonstrated hepatoprotective, antidiabetic, and lipid metabolism-modulating effects [39,40]. Meanwhile, theobromine, a methylxanthine alkaloid with milder stimulant effects than caffeine, exhibits vasodilatory, diuretic, and antioxidant activities, thereby enhancing the functional value of the *vinal* extract [41]. The theobromine concentrations determined in Neltuma ruscifolia extracts, particularly in the one obtained through DME (74.9 µg/mL), significantly exceed the levels reported in traditional beverages, such as chocolate (17–21 µg/mL), coffee (12–17 µg/mL), and tea (9–21 µg/mL), as reported by Bispo et al. [42]. This suggests that *N. ruscifolia* extracts could represent a relevant alternative source of theobromine with functional and nutraceutical potential. Additionally, caffeic acid is also recognized for its antioxidant, anti-inflammatory, and antimicrobial effects [43]. Furthermore, rutin, a flavonoid known for stabilizing capillary walls, offers vasoprotective, antioxidant, and anti-inflammatory properties, making it particularly relevant in nutraceutical and dermocosmetic formulations [44]. Finally, although present in low concentrations, cinnamic acid contributes antimicrobial and antioxidant properties, complementing the extract’s bioactive profile through synergistic effects [45].

Previous studies have reported differences in the polyphenolic profiles of *Prosopis alba* and *Prosopis nigra*, with quercetin, apigenin, kaempferol, and isovitexin as the predominant compounds, and *P. nigra* exhibiting greater chemical diversity than *P. alba* [34]. Other studies have shown that rutin and quercetin are the main polyphenols in *Prosopis laevigata* [12]. As previously mentioned, *Neltuma ruscifolia* contains gallic acid, rutin, and theobromine, further highlighting the chemical diversity and potential bioactivity across *Neltuma* (syn. *Prosopis*) species.

#### 3.2.4. Principal Component Analysis (PCA)

The analysis accounted for 99% of the total variability through the first two components (Figure 6). The first principal component (PC1), which explains 79% of the variance, can be interpreted as a functional differentiation axis, as it distinguishes the extraction methods based on their bioactive compound content. Along this axis, DME is positioned at the positive end, associated with higher concentrations of phenolic compounds identified by HPLC (caffeic acid, theobromine, rutin, and cinnamic acid), as well as higher TPC, AC, and TEAC (DPPH and ABTS assays). In contrast, UE and ME are grouped on the negative end of this axis, indicating a lower yield in the extraction of these functional compounds.

The second component (PC2), explaining the remaining 20.0% of the variability, can be interpreted as an axis of differentiation in terms of energy efficiency. On this axis, ME shows a positive association with higher REE, while UE appears on the negative end and is more closely associated with TF. This distribution suggests an inverse relationship between REE and TF content, indicating that methods with lower energy demand, such as UE, promote the extraction of flavonoids, whereas energy-intensive techniques like ME are less effective in this regard. Thus, PCA effectively differentiates the extraction methods based on both their capacity to recover bioactive compounds and their energetic performance.

Although energy efficiency is an important consideration in the development of sustainable processes, in this case, it takes a secondary role to the primary goal of obtaining extracts with high functional value and elevated concentrations of polyphenols, both total and individual. This is particularly relevant in applications within the food, pharmaceutical, or cosmetic industries, where antioxidant potential is a key functional attribute.

In this regard, despite its superior REE, UE proved to be the least effective method for the extraction of bioactive compounds. Consequently, achieving comparable levels of functionality would require a greater amount of raw material. The method’s excellent energy performance stems mainly from its low energy consumption; however, this advantage is offset by its limited extraction capacity.

From a broader perspective, considering the quantity of *vinal* needed to obtain functionally equivalent extracts, UE emerges as the least favorable option. Its large-scale implementation would entail higher demands for cultivation area, labor for harvesting and processing, storage capacity, and infrastructure, ultimately resulting in increased production costs.

### 3.3. Selection of the Optimal Method for Obtaining Functional Extracts

The results from the analyses applied in the functional characterization of the extracts were considered to select the optimal extraction technique. Initially, the optimized extracts were compared against TME to contextualize the improvements achieved. In all evaluated parameters (TPC, TF, AC, and TEAC), DME showed significantly higher values than TME (*p* < 0.05), positioning it as the most effective technique for enhancing extract functionality. In particular, a marked increase in AC and TPC was observed, which is crucial for applications aiming to maximize bioactive potential. While ME and UE outperformed the traditional method in some respects, their overall performance was inferior to DME. According to Zhou et al. [46], the reduced efficiency in polyphenol extraction and AC during UE in certain plant matrices may result from the chemical degradation of specific polyphenols at amplitudes exceeding 21.25%, combined with thermal degradation due to excessive ultrasonic pulses. Similarly, Radnia et al. [47] noted that high ME power (e.g., >200 W) can cause denaturation of polyphenolic compounds, leading to a reduction in their antioxidant properties.

Additionally, the phenolic profile determined by HPLC reinforces DME superiority. This method yielded significantly higher concentrations of the major compounds, especially gallic acid, rutin, and theobromine.

The principal component analysis provided a comprehensive synthesis of the analytical and energetic information, offering a clear differentiation among treatments. DME was positioned at the positive end of the main component associated with functional and antioxidant content, consolidating its status as the most effective method. Although UE stood out for its relative energy efficiency, this advantage was offset by its poor performance in extracting functional compounds, making it a less suitable option overall.

Taken together, these findings support DME as the best alternative for obtaining functional *vinal* extracts. It ensures a high content of bioactive compounds and excellent AC while also offering practical and economic advantages, as it does not require specialized infrastructure, making it an accessible option for implementation at a productive scale without compromising extractive quality.

## 4. Conclusions

DME, optimized for 40 min, 50 °C, and 42% ethanol, proved to be the most efficient method, compared to ME and UE, achieving the highest total phenolic compound contents and requiring the lowest solvent volumes. Furthermore, DME proved to be technically and economically viable, as well as environmentally sustainable, for scale-up.The CA of the optimized *vinal* extracts was up to four times higher than that of tea, coffee, and berries.The application of RSM was crucial to optimizing these methods, allowing not only the identification of optimal operating conditions but also revealing complex interactions between the factors studied. This interrelationship demonstrates that process efficiency depends on a delicate balance between the applied conditions.To the best of our knowledge, this study represents the first documented comparison of extraction techniques for optimizing the extraction of bioactive compounds from *Neltuma* spp., employing RSM, and considering energy and carbon footprint.This work underscores the importance of integrating advanced statistical tools with functional characteristics and energy, environmental, and practical criteria to develop more efficient, affordable, and adaptable extraction processes.Favoring the added value of native species also drives the development of bioactive ingredients for innovative applications and sustainable industrial scaling.

## Figures and Tables

**Figure 1 foods-14-02927-f001:**
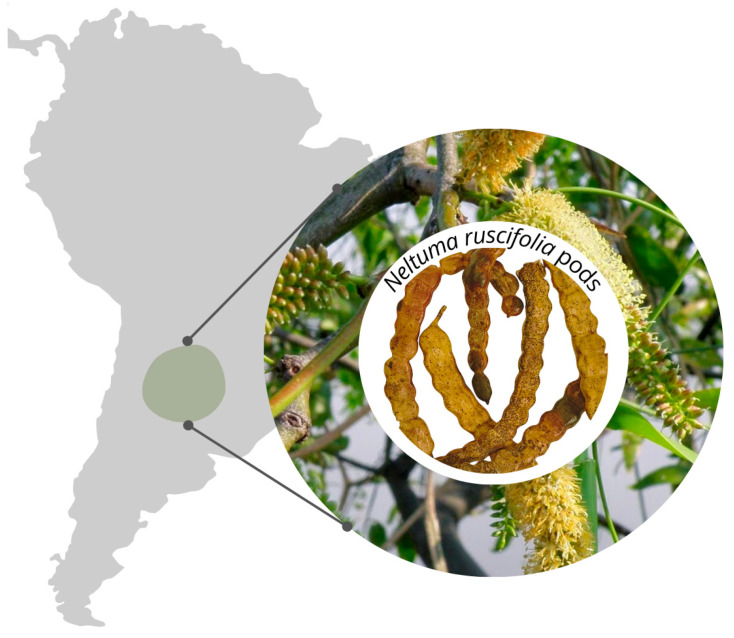
*Neltuma ruscifolia* (*vinal*) pods, branches, and flowers with the area of occurrence in Argentina, South America.

**Figure 2 foods-14-02927-f002:**
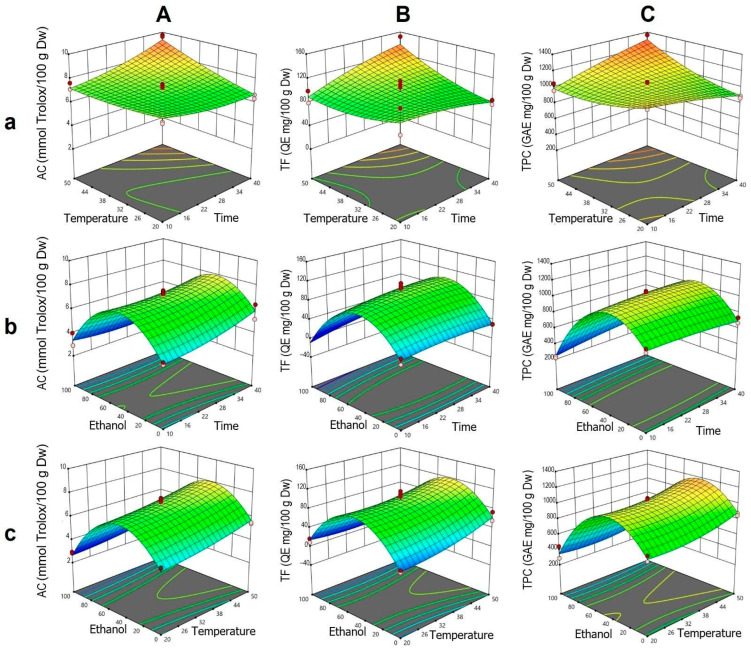
Response surface plots showing the optimization of *vinal* extract production by dynamic maceration. Each plot represents the effect of factor combinations (lowercase letters), namely, temperature vs. time (**a**), ethanol concentration vs. time (**b**), and ethanol concentration vs. temperature (**c**), on the responses (uppercase letters): antioxidant capacity (AC-DPPH assay-) (**A**), total flavonoid content (**B**), and total polyphenolic content (**C**). Red dots indicate the optimal extraction conditions.

**Figure 3 foods-14-02927-f003:**
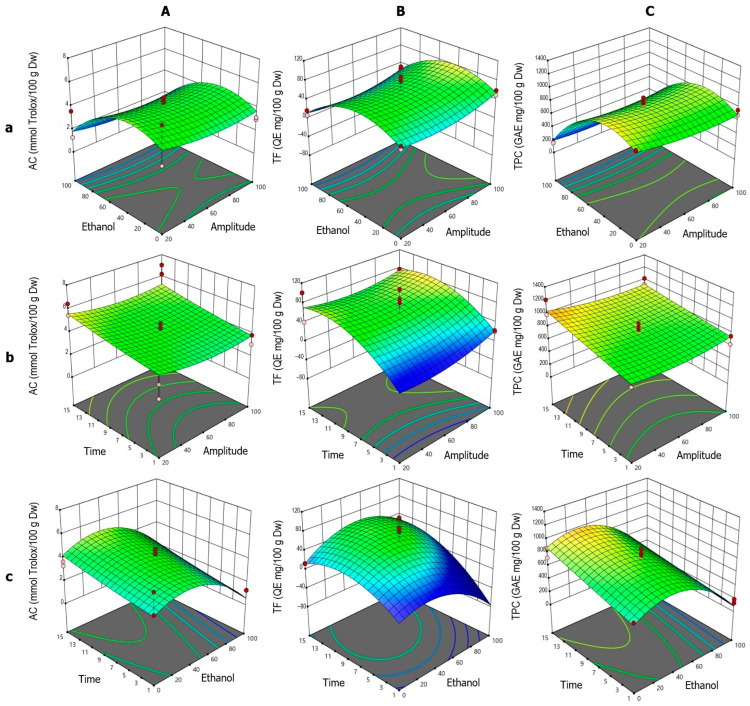
Response surface plots showing the optimization of *vinal* extract production by ultrasound-assisted extraction. Each plot represents the effect of factor combinations (lowercase letters), namely, ethanol concentration vs. amplitude (**a**), time vs. amplitude (**b**), and time vs. ethanol concentration (**c**), on the responses (uppercase letters): antioxidant capacity (AC-DPPH assay-) (**A**), total flavonoid content (**B**), and total polyphenolic content (**C**). Red dots indicate the optimal extraction conditions.

**Figure 4 foods-14-02927-f004:**
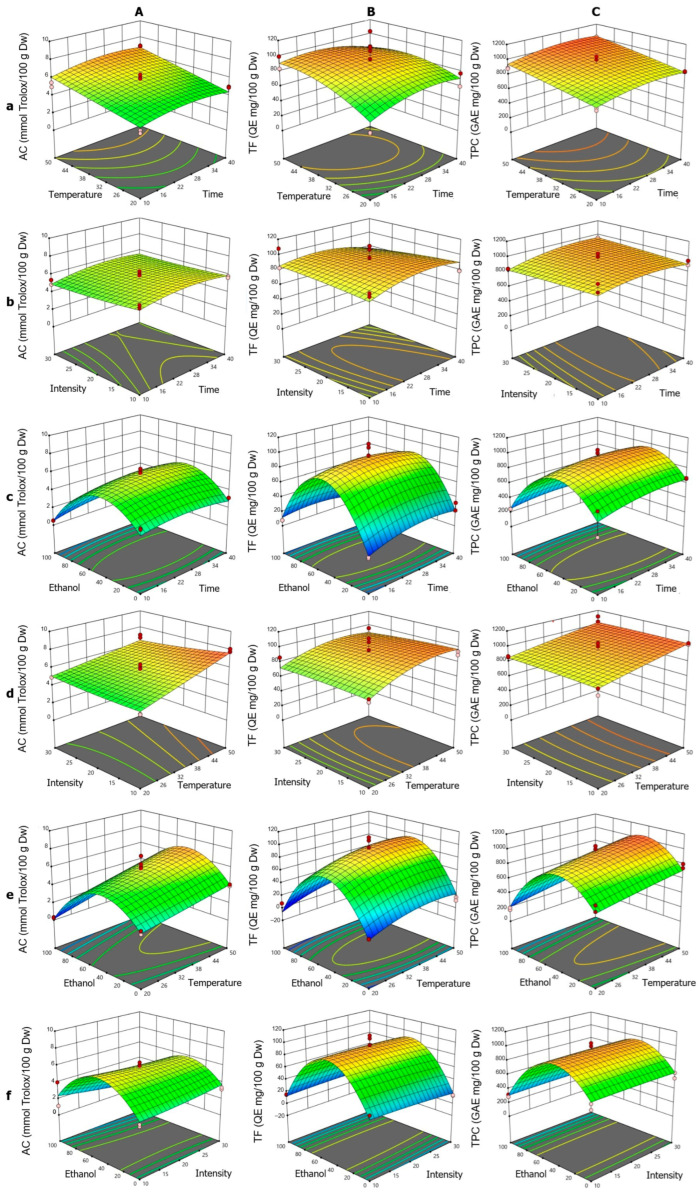
Response surface plots showing the optimization of *vinal* extract production by microwave-assisted extraction. Each plot represents the effect of factor combinations (lowercase letters), namely, temperature vs. time (**a**), intensity vs. time (**b**), ethanol concentration vs. time (**c**), intensity vs. temperature (**d**), ethanol concentration vs. temperature (**e**), and ethanol concentration vs. intensity (**f**), on the responses (uppercase letters): antioxidant capacity (AC-DPPH assay-) (**A**), total flavonoid content (**B**), and total polyphenolic content (**C**). Red dots indicate the optimal extraction conditions.

**Figure 5 foods-14-02927-f005:**
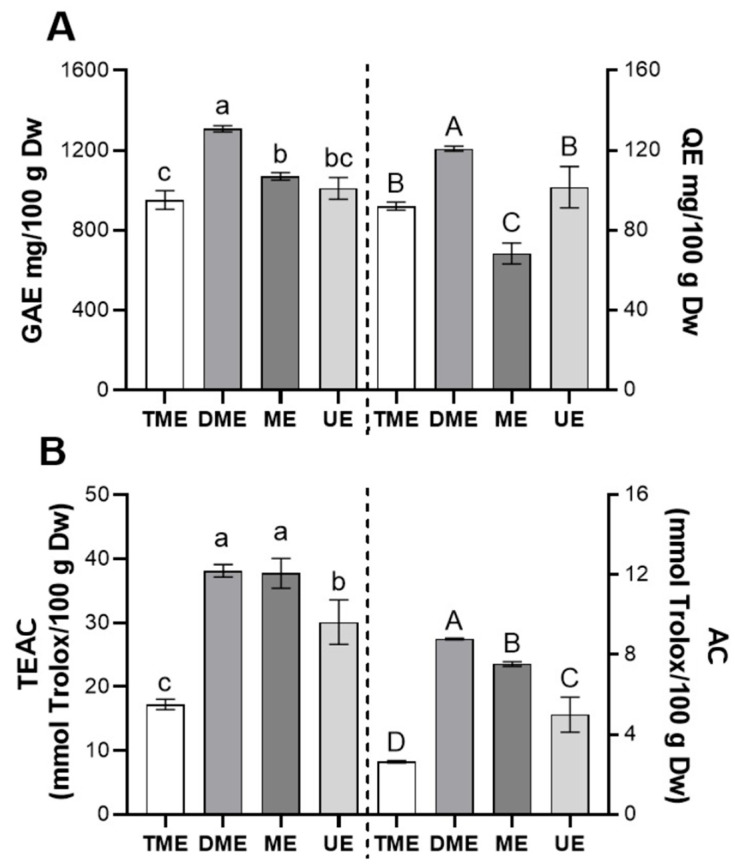
Comparison of the bioactivity of optimal *vinal* extracts obtained by different extraction methods. (**A**) Total polyphenols, expressed as mg of gallic acid per gram (*vinal* pod flour) on a dry weight basis (Dw), and total flavonoids, expressed as mg of quercetin per g Dw; (**B**) scavenging activity determined by the TEAC and AC assays, expressed as mmol Trolox per gram Dw. Extraction methods: traditional maceration (TME), dynamic maceration (DME), microwave-assisted extraction (ME), and ultrasound-assisted extraction (UE). Different letters indicate significant differences between extraction treatments (lowercase letters on the left; uppercase letters on the right) (Tukey’s test, *p* < 0.05).

**Figure 6 foods-14-02927-f006:**
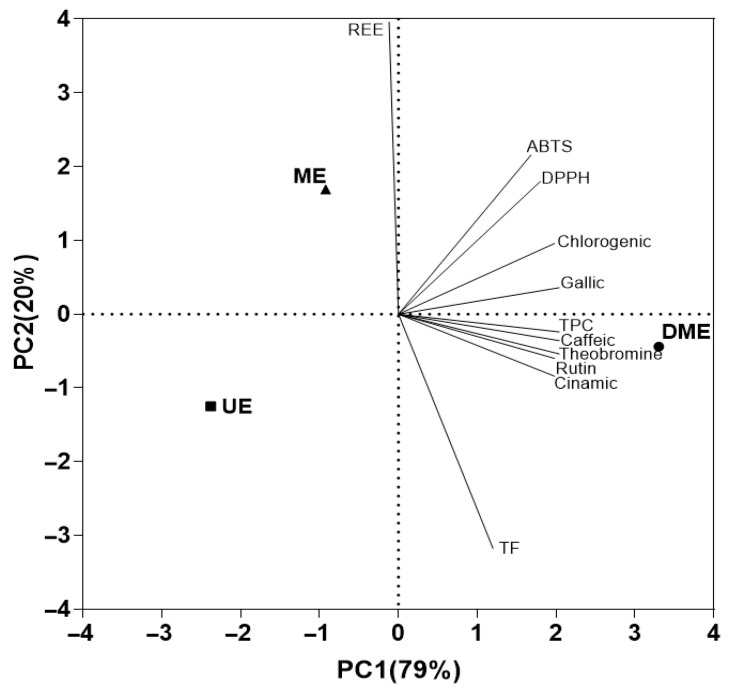
Principal component analysis (PCA) plot showing the relationships among the measured responses in the extracts, their chromatic characteristics, and the relevant wavelengths.

**Table 1 foods-14-02927-t001:** Phenolic compounds identified and quantified (ppm) by HPLC in *vinal* extracts. The table presents integration wavelengths, compound concentrations for each extraction method (dynamic maceration—DME, microwave-assisted extraction—ME, and ultrasound-assisted extraction—UE), and percentage yields (for ME and UE) relative to DME. Different letters within the same row indicate significant differences between mean values (*p* < 0.05).

Compounds	Wavelength (nm)	DME	ME		UE	
Gallic acid	210	168 ± 5 ^a^	60 ± 4 ^b^	35.8%	3 ± 1 ^c^	1.9%
Chlorogenic	325	5 ± 1 ^a^	3 ± 1 ^b^	68.7%	2 ± 1 ^c^	40.8%
Theobromine	271	75 ± 3 ^a^	20 ± 1 ^b^	26.5%	12 ± 1 ^c^	16.5%
Caffeic	320	6 ± 1 ^a^	3 ± 1 ^b^	51.0%	2 ± 1 ^c^	40.5%
Rutin	355	114 ± 8 ^a^	27 ± 5 ^b^	23.8%	17 ± 1 ^c^	14.8%
Cinnamic	275	7 ± 1 ^a^	4 ± 1 ^b^	52.8%	4 ± 1 ^b^	51.0%

## Data Availability

The data are available at https://drive.google.com/drive/folders/1G6GnaqHqfpMGQgQ4u7i2Tz83AI4gzzcb?usp=drive_link (accesed on 1 March 2025).

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
