# Peer review of "Integrated Approach for the Optimization of the Sustainable Extraction of Polyphenols from a South American Abundant Edible Plant: Neltuma ruscifolia"

_foods, 2025, doi:10.3390/foods14172927_

Round 1

Reviewer 1 Report

Comments and Suggestions for Authors

In the article entitled "Integrated approach for the optimization of the sustainable extraction of polyphenols from a South American abundant edible plant: Neltuma ruscifolia," the authors address an important issue related to the use of neglected edible plants in the context of obtaining bioactive compounds. The authors present an integrated approach to the optimization of three polyphenol extraction methods, emphasizing energy efficiency and sustainability. Despite its high substantive value, the work contains several elements that require refinement in terms of clarity and structure.

Many sentences in the abstract are very long and complex, making the information difficult to understand. They should be simplified and shortened for greater clarity. The abstract lacks logical divisions such as background/problem, purpose, methods, results, and conclusions. Information is jumbled, for example, results appear before a clear description of the methods. Data on DPPH, ABTS, TPC, TF, and energy consumption, among others, are important, but in the abstract, it's worth limiting them to the most representative results rather than providing a full range of values.

Language errors should be corrected, e.g. the phrase "like for many other" is stylistically awkward - it should be "As with many other"; The construction "in a secondary place" is awkward and stylistically incorrect. More correct: and, to a lesser extent, by the extraction method. "The pods powder was obtained..." - grammatical error – incorrect noun construction. Correct: The pod powder was obtained... or Powder from the pods was obtained... "Brieftly" - spelling error – should be: Briefly.
The entire work requires linguistic proofreading!

The introduction should be updated and based on newer scientific sources.

The aim of the study was not clearly and specifically formulated. The current version is verbose and conceals the research objective within a complex structure.

All units except degrees Celsius and % should be separated from the number by a space.

In their discussion of the results, the authors provide only numerical values, without any deeper analysis - e.g. whether the differences are statistically significant, what mechanisms may be behind such results.

Table 1 should be included in the appendix. It is difficult to read and adds little value when placed here in the paper.

The conclusions are consistent with the study's objectives – they address the optimization of extraction methods, the effectiveness of dynamic maceration, and the potential of Neltuma ruscifolia as a source of bioactive compounds. Conclusions should be concise and concise, although some information is repeated several times, for example:
- "Dynamic maceration... achieved the highest contents of phenolic compounds and flavonoids..." (repeated in two paragraphs);
- "response surface methodology..." – conclusions regarding RSM can be shortened to one sentence.
I suggest limiting conclusions to 3-5 main points without duplication.

Author Response

foods-3810822

Integrated approach for the optimization of the sustainable extraction of polyphenols from a South American abundant edible plant: Neltuma ruscifolia

Response to reviewers

REVIEWER 1 

In the article entitled "Integrated approach for the optimization of the sustainable extraction of polyphenols from a South American abundant edible plant: Neltuma ruscifolia," the authors address an important issue related to the use of neglected edible plants in the context of obtaining bioactive compounds. The authors present an integrated approach to the optimization of three polyphenol extraction methods, emphasizing energy efficiency and sustainability. Despite its high substantive value, the work contains several elements that require refinement in terms of clarity and structure.

Many sentences in the abstract are very long and complex, making the information difficult to understand. They should be simplified and shortened for greater clarity. The abstract lacks logical divisions such as background/problem, purpose, methods, results, and conclusions. Information is jumbled, for example, results appear before a clear description of the methods. Data on DPPH, ABTS, TPC, TF, and energy consumption, among others, are important, but in the abstract, it's worth limiting them to the most representative results rather than providing a full range of values.

R: The abstract has been rewritten taking the reviewer´s suggestions into account. 

Language errors should be corrected, e.g. the phrase "like for many other" is stylistically awkward - it should be "As with many other"; The construction "in a secondary place" is awkward and stylistically incorrect. More correct: and, to a lesser extent, by the extraction method. "The pods powder was obtained..." - grammatical error – incorrect noun construction. Correct: The pod powder was obtained... or Powder from the pods was obtained... "Brieftly" - spelling error – should be: Briefly.
The entire work requires linguistic proofreading!

R: The whole manuscript was revised for grammatical errors and linguistic proofreading. 

The introduction should be updated and based on newer scientific sources.

R: The introduction was updated with newer references. 

The aim of the study was not clearly and specifically formulated. The current version is verbose and conceals the research objective within a complex structure.

R: The aim of the study was justified and clarified. 

All units except degrees Celsius and % should be separated from the number by a space.

R: We have revised the manuscript to assess that all units, except for degrees Celsius (°C) and percentages (%), are now separated from the numerical values by a space, in accordance with journal formatting guideline.

In their discussion of the results, the authors provide only numerical values, without any deeper analysis - e.g. whether the differences are statistically significant, what mechanisms may be behind such results.

R: A deeper discussion of the obtained results has been performed  

Table 1 should be included in the appendix. It is difficult to read and adds little value when placed here in the paper.

R: Table 1 has been moved to the appendix as Table S2.

The conclusions are consistent with the study's objectives – they address the optimization of extraction methods, the effectiveness of dynamic maceration, and the potential of Neltuma ruscifolia as a source of bioactive compounds. Conclusions should be concise and concise, although some information is repeated several times, for example:
- "Dynamic maceration... achieved the highest contents of phenolic compounds and flavonoids..." (repeated in two paragraphs);
- "response surface methodology..." – conclusions regarding RSM can be shortened to one sentence.
I suggest limiting conclusions to 3-5 main points without duplication.

R: We appreciate the reviewer’s constructive feedback regarding the conciseness of the conclusions. In the revised version, we have streamlined the section to avoid repetition and limited it to four main points directly addressing the study’s objectives. The repeated statements on dynamic maceration and response surface methodology have been merged into single, concise sentences. The numbering has been corrected, and the conclusions now emphasize only the most significant findings.

Reviewer 2 Report

Comments and Suggestions for Authors

The authors present a comparative study evaluating three extraction methods for polyphenols from Neltuma ruscifolia, assessing yield, antioxidant activity, flavonoids, energy consumption, and carbon footprint. The work is systematic and provides valuable data. However, significant revisions are required to improve clarity, accuracy, and adherence to journal standards before acceptance.

1 Abstract: The abstract (305 words) exceeds typical length limits and should be condensed. Focus on key objectives, methods, results, and conclusions concisely.

2 Section 2.4.1 (Lines 141–152) duplicates text from Lines 129–140. Remove the repetition and ensure all methodological descriptions are unique and succinct.

3 Table 1 deviates from journal formatting guidelines. Revise all tables (e.g., font, alignment, units) to comply with the prescribed style.

4 Several editorial issues require correction throughout the manuscript:

  1. Latin Names: Italicize species names (Neltuma ruscifolia, Prosopis nigra) consistently.
  2. Typos & Errors: Correct pervasive typographical errors (e.g., “Figureure,” “XX MPa,” “ranging from XX to XX”).
  3. Abbreviations: Define abbreviations (e.g., TPC) at first use and ensure consistency. The full term for "RCF" is missing entirely.

5 Kindly improve on the Results and Discussion. The overall structure needs to be improved. Many large paragraphs and sentences, kindly break them for better understanding.

6 Conclusion: The conclusion is erroneously labeled as “5” (should be “4”). Revise numbering and streamline content to emphasize only the most significant findings.

7 Supplementary material Tables S2–S7 are absent. Provide all supplementary data referenced in the manuscript.

8 Kindly refer below paper as it is highly relevant to this report: Green Extraction of Polyphenols via Deep Eutectic Solvents and Assisted Technologies from Agri-Food By-Products, Molecules, DOI10.3390/molecules28196852

Author Response

REVIEWER 2

The authors present a comparative study evaluating three extraction methods for polyphenols from Neltuma ruscifolia, assessing yield, antioxidant activity, flavonoids, energy consumption, and carbon footprint. The work is systematic and provides valuable data. However, significant revisions are required to improve clarity, accuracy, and adherence to journal standards before acceptance.

1 Abstract: The abstract (305 words) exceeds typical length limits and should be condensed. Focus on key objectives, methods, results, and conclusions concisely.

R: The abstract has been rewritten and condensed (250 words).

2 Section 2.4.1 (Lines 141–152) duplicates text from Lines 129–140. Remove the repetition and ensure all methodological descriptions are unique and succinct.

R: The duplicated text in Section 2.4.1 (Lines 141–152) has been removed, and the methodological description was revised to ensure all sections are unique, succinct, and free from repetition.

3 Table 1 deviates from journal formatting guidelines. Revise all tables (e.g., font, alignment, units) to comply with the prescribed style.

R: Table 1 has been moved to the appendix as Table S2.

4 Several editorial issues require correction throughout the manuscript:

  1. Latin Names: Italicize species names (Neltuma ruscifoliaProsopis nigra) consistently.
  2. Typos & Errors: Correct pervasive typographical errors (e.g., “Figureure,” “XX MPa,” “ranging from XX to XX”).
  3. Abbreviations: Define abbreviations (e.g., TPC) at first use and ensure consistency. The full term for "RCF" is missing entirely.

R: Thank you for the suggestions. We have corrected all editorial issues throughout the manuscript, including: italicizing Latin species names consistently, fixing typographical errors, and defining all abbreviations at first use (including RCF).

5 Kindly improve on the Results and Discussion. The overall structure needs to be improved. Many large paragraphs and sentences, kindly break them for better understanding.

R: We have revised the Results and Discussion section to improve its overall structure. Long paragraphs were split into shorter ones, and overly long sentences were restructured to enhance clarity and readability.

6 Conclusion: The conclusion is erroneously labeled as “5” (should be “4”). Revise numbering and streamline content to emphasize only the most significant findings.

R: The conclusion section numbering has been corrected (now labeled as “4”). The content has been streamlined to emphasize only the most significant findings, as suggested.

7 Supplementary material Tables S2–S7 are absent. Provide all supplementary data referenced in the manuscript.

R: References to Tables S2–S7 have been removed; only Tables S1, S2, and S3 are now included in the supplementary material.

8 Kindly refer below paper as it is highly relevant to this report: Green Extraction of Polyphenols via Deep Eutectic Solvents and Assisted Technologies from Agri-Food By-Products, Molecules, DOI10.3390/molecules28196852

R: Thank you for the suggestion. We have added the recommended reference and incorporated some of its relevant results and discussion into our manuscript.

Reviewer 3 Report

Comments and Suggestions for Authors

Q1. The Introduction lacks discussion of key aspects such as: (a) the concept of optimization, and (b) the various antioxidant compounds reported in Neltuma ruscifolia.

Q2. Lines 76–85. The section should provide a deeper discussion of the extraction methods being optimized. What results have these methods yielded in the extraction of similar compounds either from the matrix under study or from comparable matrices?

Q3. Lines 86–90. The current research needs to be better justified and its relevance more clearly emphasized. How could the optimal extracts be applied? What benefits does this research offer? What is the novelty of this study compared to previous related works?

Q4. The ANOVA tables appear to be missing. It is recommended to include them as supplementary material.

Q5. Lines 118–125. The factors and their respective levels used for the optimization of the extraction methods should be presented more clearly and systematically.

Q6. Line 129. The antioxidant activity methodology is repeated; please review and correct this duplication.

Q7. Table 1. Please include definitions for the abbreviations used in the equations at the end of the table to improve clarity and understanding.

Q8. Line 541. "XX MPA" appears to be incomplete. Please verify and complete the missing information.

Q9. Line 580. The chromatogram of the analyzed samples is missing; it should be included to support the reported results.

Q10. Line 695. The phrase “Ranging from XX to XX” is missing specific values. Please verify and complete this information.

Author Response

REVIEWER 3 

Q1. The Introduction lacks discussion of key aspects such as: (a) the concept of optimization, and (b) the various antioxidant compounds reported in Neltuma ruscifolia.

R: The Introduction has been revised to include (a) the concept of optimization, highlighting the identification of conditions that maximize bioactive compound recovery efficiently, and (b) the main antioxidant compounds reported in Neltuma ruscifolia (gallic acid, rutin, and theobromine), emphasizing their functional relevance.

Q2. Lines 76–85. The section should provide a deeper discussion of the extraction methods being optimized. What results have these methods yielded in the extraction of similar compounds either from the matrix under study or from comparable matrices? 

R: We have expanded the discussion of the extraction methods and included Table S3, which summarizes TPC results obtained using different extraction methods from similar matrices within the Prosopis/Neltuma genus. This provides context and allows comparison with our optimized extraction results for Neltuma ruscifolia.

Q3. Lines 86–90. The current research needs to be better justified and its relevance more clearly emphasized. How could the optimal extracts be applied? What benefits does this research offer? What is the novelty of this study compared to previous related works?

R: The revised manuscript now clearly justifies the relevance of the research and highlights its potential applications.

Q4. The ANOVA tables appear to be missing. It is recommended to include them as supplementary material.
R: The ANOVA tables had been incorporated into Supplementary files. 

Q5. Lines 118–125. The factors and their respective levels used for the optimization of the extraction methods should be presented more clearly and systematically.

R: The factors and their respective levels for the optimization of the extraction methods will be presented more clearly and systematically in the Supplementary Files.

Q6. Line 129. The antioxidant activity methodology is repeated; please review and correct this duplication.

R: The duplication in the description of the antioxidant activity methodology has been removed, and the section has been carefully reviewed to ensure clarity and consistency.

Q7. Table 1. Please include definitions for the abbreviations used in the equations at the end of the table to improve clarity and understanding.

R: Done

Q8. Line 541. "XX MPA" appears to be incomplete. Please verify and complete the missing information.

R: The information was added as suggested. 

Q9. Line 580. The chromatogram of the analyzed samples is missing; it should be included to support the reported results.

R: The chromatogram  has been incorporated into Supplementary files. 

Q10. Line 695. The phrase “Ranging from XX to XX” is missing specific values. Please verify and complete this information.

R: Thank you for pointing this out. The placeholder “Ranging from XX to XX” has been replaced with the correct numerical values throughout the manuscript, ensuring that all data are now accurately reported.

Reviewer 4 Report

Comments and Suggestions for Authors

General comments

The English is generally fine, but the manuscript is full of typos and basic mistakes. It needs a full proofread. It’s clear this version was not carefully read by all authors before submission.

Abstract

Avoid using acronyms and abbreviations the first time a term appears (e.g., write “total phenolic compounds” instead of “TPC”). Either remove abbreviations completely from the abstract or define them properly.

Introduction

What is the novelty here? What exactly are the authors doing differently from previous works on the valorization of Neltuma ruscilofia? Make a clear, explicit statement about this both in the abstract and the introduction.

Remove as many references older than five years as possible. References older than 2021 should not appear unless absolutely essential.

Methodology

Add a figure of the plant material so the reader can actually see what the sample looks like.

Section 2.2: Very superficial description. For DME: where was it conducted? Orbital shaker? Which rpm? Temperature? Duration? The same applies for UAE and MAE, add details about equipment, power, frequency, and duration.

Section 2.4.1: Paragraphs are duplicated. Remove the duplicated part.

Why didn’t the authors determine pigments like carotenoids, chlorophylls, or others? These are important classes of bioactive compounds that could add value.

What wavelengths were used in the HPLC/PDA analysis? This is basic info and is essential for reproducibility. If different compounds were measured at different wavelengths, list them all. Include the calibration curve range and R² for each compound.

How do the authors justify using a 2³ design for DME and UAE, but a 2⁴ for MAE? Using the same number of variables across all designs is important for fair comparison if the goal is to pick the best extraction method. A 2⁴ design also increases the number of runs, which helps statistical modeling. Explain why this was not balanced.

Another inconsistency: DME and UAE both used three variables, but not the same variables. This compromises the comparison across methods.

Why was ABTS not included as a response variable in the DoE?

Results

Line 265: What is the unit? mmol Trolox/100 g or mmol Trolox/100 mL? The abstract and methods say one thing, and here it’s another. Same confusion in lines 273, 276, and in the figures. In the surface response figures, the results are per 100 mL of extract, but elsewhere they’re per 100 g dry matter. Decide: either report concentration in extract or yield relative to dry matter, and keep it consistent. The current mix is confusing.

Table 1: Add a footnote explaining what each coded letter/variable means. Add units to the responses. And number the equations.

There are typos everywhere: “(Figure 1)”, “FIGURE” in caps, missing words, etc. Proofread everything.

The text says there’s more than one table in the Supplementary Material, but only Table S1 exists. If that info is important enough to mention in the text, put it in the main manuscript.

Section 3.2: Again, typos everywhere when writing “FIGURE”. Please fix.

Figure captions are messy and confusing. Completely redo them to be clear and correct, especially Figure 4, where the caption mentions “C”, but there are only two graphs.

Line 539: What is “XX MPA”?? This section is so poorly written that it should be completely rewritten. I had to read it multiple times to even guess what it means, and it’s still confusing. Honestly, I doubt this manuscript was reviewed by all authors before submission. Some mistakes are too obvious to be missed by five different people. The authors need to do better.

Table 2: Wrong format for the journal. Fix the format and add units for all variables and results.

Strongly recommend adding a table comparing your results with literature data, since you have three extraction methods and several response variables. It would help readers see what is new.

Conclusion

Line 693: What is “XX and XX”?? This placeholder should have been replaced with real numbers. Finding this in the first line of the conclusion shows nobody did even a quick final read.

Author Response

REVIEWER 4

General comments

The English is generally fine, but the manuscript is full of typos and basic mistakes. It needs a full proofread. It’s clear this version was not carefully read by all authors before submission.

Abstract

The authors present a comparative study evaluating three extraction methods for polyphenols from Neltuma ruscifolia, assessing yield, antioxidant activity, flavonoids, energy consumption, and carbon footprint. The work is systematic and provides valuable data. However, significant revisions are required to improve clarity, accuracy, and adherence to journal standards before acceptance.

1 Abstract: The abstract (305 words) exceeds typical length limits and should be condensed. Focus on key objectives, methods, results, and conclusions concisely.

R: The abstract has been rewritten and condensed (250 words).

2 Section 2.4.1 (Lines 141–152) duplicates text from Lines 129–140. Remove the repetition and ensure all methodological descriptions are unique and succinct.

R: The duplicated text in Section 2.4.1 (Lines 141–152) has been removed, and the methodological description was revised to ensure all sections are unique, succinct, and free from repetition.

3 Table 1 deviates from journal formatting guidelines. Revise all tables (e.g., font, alignment, units) to comply with the prescribed style.

R: Table 1 has been moved to the appendix as Table S2.

4 Several editorial issues require correction throughout the manuscript:

  1. Latin Names: Italicize species names (Neltuma ruscifoliaProsopis nigra) consistently.
  2. Typos & Errors: Correct pervasive typographical errors (e.g., “Figureure,” “XX MPa,” “ranging from XX to XX”).
  3. Abbreviations: Define abbreviations (e.g., TPC) at first use and ensure consistency. The full term for "RCF" is missing entirely.

R: Thank you for the suggestions. We have corrected all editorial issues throughout the manuscript, including: italicizing Latin species names consistently, fixing typographical errors, and defining all abbreviations at first use (including RCF).

5 Kindly improve on the Results and Discussion. The overall structure needs to be improved. Many large paragraphs and sentences, kindly break them for better understanding.

R: We have revised the Results and Discussion section to improve its overall structure. Long paragraphs were split into shorter ones, and overly long sentences were restructured to enhance clarity and readability.

6 Conclusion: The conclusion is erroneously labeled as “5” (should be “4”). Revise numbering and streamline content to emphasize only the most significant findings.

R: The conclusion section numbering has been corrected (now labeled as “4”). The content has been streamlined to emphasize only the most significant findings, as suggested.

7 Supplementary material Tables S2–S7 are absent. Provide all supplementary data referenced in the manuscript.

R: References to Tables S2–S7 have been removed; only Tables S1, S2, and S3 are now included in the supplementary material.

8 Kindly refer below paper as it is highly relevant to this report: Green Extraction of Polyphenols via Deep Eutectic Solvents and Assisted Technologies from Agri-Food By-Products, Molecules, DOI10.3390/molecules28196852

R: Thank you for the suggestion. We have added the recommended reference and incorporated some of its relevant results and discussion into our manuscript.

Avoid using acronyms and abbreviations the first time a term appears (e.g., write “total phenolic compounds” instead of “TPC”). Either remove abbreviations completely from the abstract or define them properly.

R: The abstract has been rewritten. Every abbreviation was defined the first time a term appeared.

Introduction

What is the novelty here? What exactly are the authors doing differently from previous works on the valorization of Neltuma ruscilofia? Make a clear, explicit statement about this both in the abstract and the introduction.

R: We have clarified the novelty of our study in both the abstract and the introduction.

Remove as many references older than five years as possible. References older than 2021 should not appear unless absolutely essential.

R: Thank you for the suggestion. We have updated several references with more recent sources wherever possible, while retaining older references only when absolutely essential.

Methodology

Add a figure of the plant material so the reader can actually see what the sample looks like.

R: A figure was incorporated as Figure 1 showing pod, branches and flowers, as well as area of occurrence of Neltuma ruscifolia.

Section 2.2: Very superficial description. For DME: where was it conducted? Orbital shaker? Which rpm? Temperature? Duration? The same applies for UAE and MAE, add details about equipment, power, frequency, and duration.

R:  We have revised Section 2.2 to provide a detailed description of all extraction methods. For DME, UAE, and MAE, we now include information on the equipment used, operational conditions, ensuring full reproducibility of the experiments.

Section 2.4.1: Paragraphs are duplicated. Remove the duplicated part.

R: The duplicated text in Section 2.4.1 (Lines 141–152) has been removed, and the methodological description was revised to ensure all sections are unique, succinct, and free from repetition.

Why didn’t the authors determine pigments like carotenoids, chlorophylls, or others? These are important classes of bioactive compounds that could add value.

The reviewer is right—there must certainly be many valuable compounds present in the extracts—but we decided to restrict this initial analysis to polyphenols and related compounds and functionalities. Work on carotenoids, chlorophylls, and others is in progress.

What wavelengths were used in the HPLC/PDA analysis? This is basic info and is essential for reproducibility. If different compounds were measured at different wavelengths, list them all. Include the calibration curve range and R² for each compound.

R: Thank you for the observation. The integration wavelengths for the chromatograms have been added to Table 2. As stated in the technique description, the comparison and quantification of the compounds were performed using the external standard method, with known concentrations of each standard, rather than calibration curves. Therefore, calibration curve ranges and R² values are not applicable in this case.

How do the authors justify using a 2³ design for DME and UAE, but a 2⁴ for MAE? Using the same number of variables across all designs is important for fair comparison if the goal is to pick the best extraction method. A 2⁴ design also increases the number of runs, which helps statistical modeling. Explain why this was not balanced.

Another inconsistency: DME and UAE both used three variables, but not the same variables. This compromises the comparison across methods.

R: We appreciate your observation. The difference in the number of evaluated variables was due to the technical characteristics of each equipment. In the case of microwave-assisted extraction (ME), the system used allows independent control of heating intensity (power, W) in addition to temperature, time, and ethanol concentration. In contrast, for dynamic maceration (DME) and ultrasound (UE) systems, we could only control the evaluated variables. For this reason, in DME and UE the number of factors was limited to three. Furthermore, the study's intention was not to directly compare models with equal numbers of factors, but rather to optimize each technique within the actual operational and control possibilities of each equipment, and subsequently contrast their performances under optimal conditions.

Why was ABTS not included as a response variable in the DoE?

R: Thank you for the comment. ABTS was not included as a response variable in the DoE because previous studies have shown that its results are highly correlated with those obtained using DPPH. However, ABTS was evaluated when comparing the optimal extracts obtained for each extraction technique.

Results

Line 265: What is the unit? mmol Trolox/100 g or mmol Trolox/100 mL? The abstract and methods say one thing, and here it’s another. Same confusion in lines 273, 276, and in the figures. In the surface response figures, the results are per 100 mL of extract, but elsewhere they’re per 100 g dry matter. Decide: either report concentration in extract or yield relative to dry matter, and keep it consistent. The current mix is confusing.

R: Thank you for the comment. All results have been standardized and are now consistently reported as mmol Trolox per 100 g of the original dry powder throughout the manuscript, including the text, tables, and figures.

Table 1: Add a footnote explaining what each coded letter/variable means. Add units to the responses. And number the equations.

R: Thank you for the suggestion. All requested information has been added, including explanations of each coded letter/variable, units for the responses, and numbering of the equations. This is now presented in Table S2.

There are typos everywhere: “(Figure 1)”, “FIGURE” in caps, missing words, etc. Proofread everything.

R: The manuscript has been thoroughly proofread, and all typographical errors, including inconsistencies in figure references, and missing words, have been corrected.

The text says there’s more than one table in the Supplementary Material, but only Table S1 exists. If that info is important enough to mention in the text, put it in the main manuscript.

R: Thank you for the comment. The supplementary material now includes only Tables S1 to S3. References to additional tables have been removed from the text to reflect this accurately.

Section 3.2: Again, typos everywhere when writing “FIGURE”. Please fix.

R: Done, thank you

Figure captions are messy and confusing. Completely redo them to be clear and correct, especially Figure 4, where the caption mentions “C”, but there are only two graphs.

R: All captions have been carefully revised and rewritten to ensure clarity, accuracy, and consistency with the figures. In particular, the caption for Figure 4 was corrected to match the number of graphs presented, and the reference to “C” was removed.

Line 539: What is “XX MPA”?? This section is so poorly written that it should be completely rewritten. I had to read it multiple times to even guess what it means, and it’s still confusing. Honestly, I doubt this manuscript was reviewed by all authors before submission. Some mistakes are too obvious to be missed by five different people. The authors need to do better.

R: The section 3.2.1. has been completely rewritten to improve clarity and readability.

Table 2: Wrong format for the journal. Fix the format and add units for all variables and results.

R: Table 2 format has been corrected and the missing  information was added. 

Strongly recommend adding a table comparing your results with literature data, since you have three extraction methods and several response variables. It would help readers see what is new.

R: We have added Table S3 to provide a comparison of our results with literature data.

Conclusion

Line 693: What is “XX and XX”?? This placeholder should have been replaced with real numbers. Finding this in the first line of the conclusion shows nobody did even a quick final read.

R: The placeholder ‘XX and XX’ has been removed.  The conclusion section was rewritten and the text has been carefully reviewed to ensure no other placeholders remain.

Round 2

Reviewer 1 Report

Comments and Suggestions for Authors

I accept the work in its current version. The authors have made appropriate changes.

Reviewer 2 Report

Comments and Suggestions for Authors

It can be accpeted in present form.

Reviewer 4 Report

Comments and Suggestions for Authors

Accepted. No further comments